# Using a Logarithmic Mapping to Enable Lower Discount Factors in Reinforcement Learning

**Harm van Seijen**
Microsoft Research Montréal
harm.vanseijen@microsoft.com

**Mehdi Fatemi**
Microsoft Research Montréal
mehdi.fatemi@microsoft.com

**Arash Tavakoli**
Imperial College London
a.tavakoli@imperial.ac.uk

## Abstract

In an effort to better understand the different ways in which the discount factor affects the optimization process in reinforcement learning, we designed a set of experiments to study each effect in isolation. Our analysis reveals that the common perception that poor performance of low discount factors is caused by (too) small action-gaps requires revision. We propose an alternative hypothesis that identifies the *size-difference* of the action-gap across the state-space as the primary cause. We then introduce a new method that enables more homogeneous action-gaps by mapping value estimates to a logarithmic space. We prove convergence for this method under standard assumptions and demonstrate empirically that it indeed enables lower discount factors for approximate reinforcement-learning methods. This in turn allows tackling a class of reinforcement-learning problems that are challenging to solve with traditional methods.

## 1 Introduction

In reinforcement learning (RL), the objective that one wants to optimize for is often best described as an undiscounted sum of rewards (e.g., maximizing the total score in a game) and a discount factor is merely introduced so as to avoid some of the optimization challenges that can occur when directly optimizing on an undiscounted objective [Bertsekas and Tsitsiklis, 1996]. In this scenario, the discount factor plays the role of a hyper-parameter that can be tuned to obtain a better performance on the true objective. Furthermore, for practical reasons, a policy can only be evaluated for a finite amount of time, making the effective performance metric a *finite-horizon, undiscounted objective*.[1]

To gain a better understanding of the interaction between the discount factor and a finite-horizon, undiscounted objective, we designed a number of experiments to study this relation. One surprising finding is that for some problems a low discount factor can result in better asymptotic performance, when a finite-horizon, undiscounted objective is indirectly optimized through the proxy of an *infinite-horizon, discounted sum*. This motivates us to look deeper into the effect of the discount factor on the optimization process.

We analyze why in practice the performance of low discount factors tends to fall flat when combined with function approximation, especially in tasks with long horizons. Specifically, we refute a number of common hypotheses and present a new one instead, identifying the primary culprit to be the size-

difference of the action gap (i.e., the difference between the values of the best and the second-best actions of a state) across the state-space.

Our main contribution is a new method that yields more homogeneous action-gap sizes for sparse-reward problems. This is achieved by mapping the update target to a logarithmic space and performing updates in that space instead. We prove convergence of this method under standard conditions.

Finally, we demonstrate empirically that our method achieves much better performance for low discount factors than previously possible, providing supporting evidence for our new hypothesis. Combining this with our analytical result that there exist tasks where low discount factors outperform higher ones asymptotically suggests that our method can unlock a performance on certain problems that is not achievable by contemporary RL methods.

## 2  Problem Setting

Consider a Markov decision process (MDP, [Puterman, 1994]) $M = \langle \mathcal{S}, \mathcal{A}, P, R, S_0 \rangle$, where $\mathcal{S}$ denotes the set of states, $\mathcal{A}$ the set of actions, $R$ the reward function $R : \mathcal{S} \times \mathcal{A} \times \mathcal{S} \to \mathbb{R}$, $P$ the transition probability function $P : \mathcal{S} \times \mathcal{A} \times \mathcal{S} \to [0, 1]$, and $S_0$ the starting state distribution. At each time step $t$, the agent observes state $s_t \in \mathcal{S}$ and takes action $a_t \in \mathcal{A}$. The agent observes the next state $s_{t+1}$, drawn from the transition probability distribution $P(s_t, a_t, \cdot)$, and a reward $r_t = R(s_t, a_t, s_{t+1})$. A *terminal state* is one that, once entered, terminates the interaction with the environment; mathematically, it can be interpreted as an absorbing state that transitions only to itself with a corresponding reward of 0. The behavior of an agent is defined by a policy $\pi$, which, at time step $t$, takes as input the history of states, actions, and rewards, $s_0, a_0, r_0, s_1, a_1, \ldots r_{t-1}, s_t$, and outputs a distribution over actions, in accordance to which action $a_t$ is selected. If action $a_t$ only depends on the current state $s_t$, we will call the policy a *stationary* one; if the policy depends on more than the current state $s_t$, we will call the policy *non-stationary*.

We define a *task* to be the combination of an MDP $M$ and a *performance metric* $F$. The metric $F$ is a function that takes as input a policy $\pi$ and outputs a score that represents the performance of $\pi$ on $M$. By contrast, we define the *learning metric* $F_l$ to be the metric that the agent optimizes. Within the context of this paper, unless otherwise stated, the performance metric $F$ considers the expected, finite-horizon, undiscounted sum of rewards over the start-state distribution; the learning metric $F_l$ considers the expected, infinite-horizon, discounted sum of rewards:

$$F(\pi, M) = \mathbb{E}\left[\sum_{i=0}^{h-1} r_i \middle| \pi, M\right] \qquad ; \qquad F_l(\pi, M) = \mathbb{E}\left[\sum_{i=0}^{\infty} \gamma^i r_i \middle| \pi, M\right] , \qquad (1)$$

where the horizon $h$ and the discount factor $\gamma$ are hyper-parameters of $F$ and $F_l$, respectively.

The optimal policy of a task, $\pi^*$, is the policy that maximizes the metric $F$ on the MDP $M$. Note that in general $\pi^*$ will be a non-stationary policy. In particular, the optimal policy depends besides the current state on the time step. We denote the policy that is optimal w.r.t. the learning metric $F_l$ by $\pi_l^*$. Because $F_l$ is not a finite-horizon objective, there exists a stationary, optimal policy for it, considerably simplifying the learning problem.[2] Due to the difference between the learning and performance metrics, the policy that is optimal w.r.t. the learning metric does not need to be optimal w.r.t. the performance metric. We call the difference in performance between $\pi_l^*$ and $\pi^*$, as measured by $F$, the *metric gap*:

$$\Delta_F = F(\pi^*, M) - F(\pi_l^*, M)$$

The relation between $\gamma$ and the metric gap will be analyzed in Section 3.1.

We consider model-free, value-based methods. These are methods that aim to find a good policy by iteratively improving an estimate of the optimal action-value function $Q^*$, which, generally, predicts the expected discounted sum of rewards under the optimal policy $\pi_l^*$ conditioned on state-action pairs. The canonical example is Q-learning [Watkins and Dayan, 1992], which updates its estimates as follows:

$$Q_{t+1}(s_t, a_t) := (1 - \alpha)Q_t(s_t, a_t) + \alpha\left(r_t + \gamma \max_{a'} Q_t(s_{t+1}, a')\right) , \qquad (2)$$

where $\alpha \in [0, 1]$ is the step-size. The action-value function is commonly estimated using a function approximator with weight vector $\theta$: $Q(s, a; \theta)$. Deep Q-Networks (DQN) [Mnih et al., 2015] use a deep neural network as function approximator and iteratively improve an estimate of $Q^*$ by minimizing a sequence of loss functions:

$$\mathcal{L}_i(\theta_i) = \mathbb{E}_{s,a,r,s'}[(y_i^{DQN} - Q(s, a; \theta_i))^2], \tag{3}$$

$$\text{with} \qquad y_i^{DQN} = r + \gamma \max_{a'} Q(s', a'; \theta_{i-1}), \tag{4}$$

The weight vector from the previous iteration, $\theta_{i-1}$, is encoded using a separate target network.

## 3 Analysis of Discount Factor Effects

### 3.1 Effect on Metric Gap

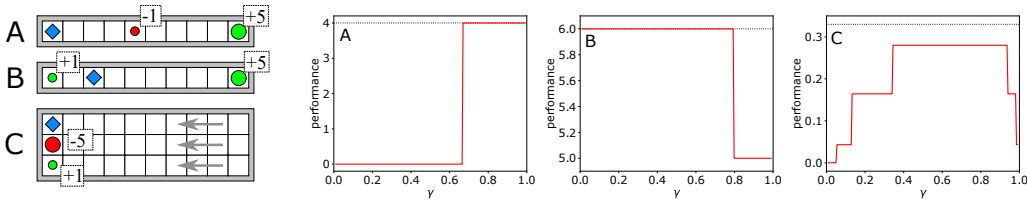

Figure 1: Illustrations of three different tasks (blue diamond: starting position; green circle: positive object; red circle: negative object; gray arrows: wind direction; numbers indicate rewards). The graphs show the performance—as measured by $F$—on these tasks for $\pi^*$ (black, dotted line) and $\pi_l^*$ (red, solid line) as function of the discount factor of the learning metric. The difference between the two represents the metric gap.

The question that is central to this section is the following: given a finite-horizon, undiscounted performance metric, what can be said about the relation between the discount factor of the learning metric and the metric gap?

To study this problem, we designed a variety of different tasks and measured the dependence between the metric gap and the discount factor. In Figure 1, we illustrate three of those tasks, as well as the metric gap on those tasks as function of the discount factor. In each task, an agent, starting from a particular position, has to collect rewards by collecting the positive objects while avoiding the negative objects. The transition dynamics of tasks A and B is deterministic; whereas, in task C wind blows in the direction of the arrows, making the agent move towards left with a 40% chance, regardless of its performed action. For all three tasks, the horizon of the performance metric is 12.

On task A, where a small negative reward has to be traded off for a large positive reward that is received later, high discount factors result in a smaller metric gap. By contrast, on task B, low discount factors result in a smaller metric gap. The reason is that for high discount factors the optimal learning policy takes the longer route by first trying to collect the large object, before going to the small object. However, with a performance metric horizon of 12, there is not enough time to take the long route and get both rewards. The low discount factor takes a shorter route by first going to the smaller object and is able to collect all objects in time. On task C, a trade-off has to be made between the risk of falling into the negative object (due to domain stochasticity) versus taking a longer detour that minimizes this risk. On this task, the optimal policy $\pi^*$ is non-stationary (the optimal action depends on the time step). However, because the learning objective $F_l$ is not finite-horizon, it has a stationary optimal policy $\pi_l^*$. Hence, the metric gap cannot be reduced to 0 for any value of the discount factor. The best discount factor is something that is not too high nor too low.

While the policy $\pi_l^*$ is derived from an infinite-horizon metric, this does not preclude it from being learned with finite-length training episodes. As an example, consider using Q-learning to learn $\pi_l^*$ for any of the tasks from Figure 1. With a uniformly random behavior policy and training episodes of length 12 (the same as the horizon of the performance metric), there is a non-zero probability for each state-action pair that it will be visited within an episode. Hence, with the right step-size decay schedule, convergence in the limit can be guaranteed [Jaakkola et al., 1994]. A key detail to enable

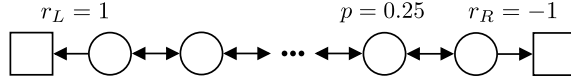

$r_L = 1$        $p = 0.25$    $r_R = -1$

Figure 2: Chain task consisting of 50 states and two terminal ones. Each (non-terminal) state has two actions: $a_L$ which results in transitioning to the left with probability $1 - p$ and to the right with probability $p$, and vice versa for the other action, $a_R$. All rewards are 0, except for transitioning to the far-left or far-right terminal states that result in $r_L$ and $r_R$, respectively.

this is that the state that is reached at the final time step is not treated as a terminal state (which has value 0 by default), but normal bootstrapping occurs [Pardo et al., 2018].

A finite-horizon performance metric is not essential to observe strong dependence of the metric gap on $\gamma$. For example, if on task B the performance metric would measure the number of steps it takes to collect all objects, a similar graph is obtained. In general, the examples in this section demonstrate that the best discount factor is task-dependent and can be anywhere in the range between 0 and 1.

## 3.2 Optimization Effects

The performance of $\pi_l^*$ gives the theoretical limit of what the agent can achieve given its learning metric. However, the discount factor also affects the optimization process; for some discount factors, finding $\pi_l^*$ could be more challenging than for others. In this section, using the task shown in Figure 2, we evaluate the correlation between the discount factor and how hard it could be to find $\pi_l^*$. It is easy to see that the policy that always takes the left action $a_L$ maximizes both discounted and undiscounted sum of rewards for any discount factor or horizon value, respectively. We define the learning metric $F_l$ as before (1), but use a different performance metric $F$. Specifically, we define $F$ to be 1 if the policy takes $a_L$ in every state, and 0 otherwise. The metric gap for this setting of $F$ and $F_l$ is 0, with the optimal performance (for $\pi^*$ and $\pi_l^*$) being 1.

To study the optimization effects under function approximation, we use linear function approximation with features constructed by tile-coding [Sutton, 1996], using tile-widths of 1, 2, 3, and 5. A tile-width of $w$ corresponds to a binary feature that is non-zero for $w$ neighbouring states and zero for the remaining ones. The number and offset of the tilings are such that any value function can be represented. Hence, error-free reconstruction of the optimal action-value function is possible in principle, for any discount factor. Note that for a width of 1, the representation reduces to a tabular one.

To keep the experiment as simple as possible, we remove exploration effects by performing update sweeps over the entire state-action space (using a step-size of 0.001) and measure performance at the end of each update sweep. Figure 3 shows the performance during early learning (average performance over the first $10,000$ sweeps) as well as the final performance (average between sweeps $100,000$ and $110,000$).

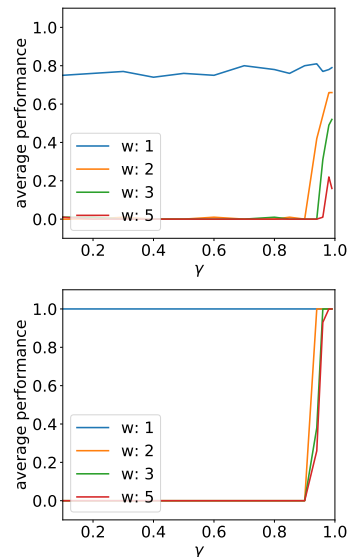

Figure 3: Early performance (top) and final performance (bottom) on the chain task.

These experiments demonstrate a common empirical observation: when using function approximation, low discount factors do not work well in sparse-reward domains. More specifically, the main observations are: 1) there is a sharp drop in final performance for discount factors below some threshold; 2) this threshold value depends on the tile-width, with larger ones resulting in worse (i.e., higher) threshold values; and 3) the tabular representation performs well for all discount factors.

It is commonly believed that the *action gap* has a strong influence on the optimization process [Bellemare et al., 2016, Farahmand, 2011]. The action gap of a state $s$ is defined as the difference in $Q^*$ between the best and the second best actions at that state. To examine this common belief, we start by evaluating two straightforward hypotheses involving the action gap: 1) lower discount factors cause poor performance because they result in smaller action gaps; 2) lower discount factors cause poor performance because they result in smaller relative action gaps (i.e, the action gap of a

state divided by the maximum action-value of that state). Since both hypotheses are supported by the results from Figure 3, we performed more experiments to test them. To test the first hypothesis, we performed the same experiment as above, but with rewards that are a factor 100 larger. This in turn increases the action gaps by a factor 100 as well. Hence, to validate the first hypothesis, this change should improve (i.e., lower) the threshold value where the performance falls flat. To test the second hypothesis, we pushed all action-values up by 100 through additional rewards, reducing the relative action-gap. Hence, to validate the second hypothesis, performance should degrade for this variation. However, neither of the modifications caused significant changes to the early or final performance, invalidating these hypotheses. The corresponding graphs can be found in the supplementary material.

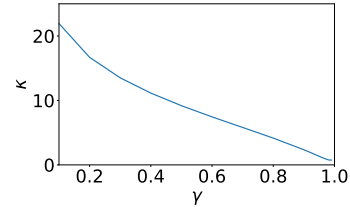

Because our two naïve action-gap hypotheses have failed, we propose an alternative hypothesis: lower discount factors cause poor performance because they result in a larger difference in the action-gap sizes across the state-space. To illustrate the statement about the difference in action-gap sizes, we define a metric, which we call the *action-gap deviation* $\kappa$, that aims to capture the notion of action-gap variations. Specifically, let $X$ be a random variable and let $\mathcal{S}^+ \subseteq \mathcal{S}$ be the subset of states that have a non-zero action gap. $X$ draws uniformly at random a state $s \in \mathcal{S}^+$ and outputs $\log_{10}(AG(s))$, where $AG(s)$ is the action gap of state $s$. We now define $\kappa$ to be the standard deviation of the variable $X$. Figure 4 plots $\kappa$ as function of the discount factor for the task in Figure 2.

Figure 4: Action-gap deviation as function of discount factor.

To test this new hypothesis, we have to develop a method that reduces the action-gap deviation $\kappa$ for low discount factors, without changing the optimal policy. We do so in the next section.

# 4 Logarithmic Q-learning

In this section, we introduce our new method, logarithmic Q-learning, which reduces the action-gap deviation $\kappa$ for sparse-reward domains. We present the method in three steps, in each step adding a layer of complexity in order to extend the generality of the method. In the supplementary material, we prove convergence of the method in its most general form. As the first step, we now consider domains with deterministic dynamics and rewards that are either positive or zero.

## 4.1 Deterministic Domains with Positive Rewards

Our method is based on the same general approach as used by Pohlen et al. [2018]: mapping the update target to a different space and performing updates in that space instead. We indicate the mapping function by $f$, and its inverse by $f^{-1}$. Values in the mapping space are updated as follows:

$$\widetilde{Q}_{t+1}(s_t, a_t) := (1-\alpha)\widetilde{Q}_t(s_t, a_t) + \alpha f\left(r_t + \gamma \max_{a'} f^{-1}\left(\widetilde{Q}_t(s_{t+1}, a')\right)\right). \tag{5}$$

Note that $\widetilde{Q}$ in this equation is not an estimate of an expected return; it is an estimate of an expected return mapped to a different space. To obtain a regular Q-value the inverse mapping has to be applied to $\widetilde{Q}$. Because the updates occur in the mapping space, $\kappa$ is now measured w.r.t. $\widetilde{Q}$. That is, the action gap of state $s$ is now defined in the mapping space as $\widetilde{Q}(s, a_{best}) - \widetilde{Q}(s, a_{2nd\,best})$.

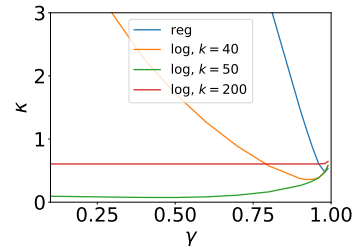

To reduce $\kappa$, we propose to use a logarithmic mapping function. Specifically, we propose the following mapping function:

$$f(x) := c \ln(x + \gamma^k) + d, \tag{6}$$

with inverse function: $f^{-1}(x) = e^{(x-d)/c} - \gamma^k$, where $c$, $d$, and $k$ are mapping hyper-parameters.

To understand the effect of (6) on $\kappa$, we plot $\kappa$, based on action gaps in the logarithmic space, on a variation of the chain task (Figure 2) that uses $r_R = 0$ and $p = 0$. We also plot $\kappa$ based on actions in the regular space. Figure 5 shows that with an appropriate value of $k$, the action-gap deviation can almost be reduced to 0 for low values of $\gamma$. Setting $k$ too high increases the deviation a little, while setting it too low increases it a lot for

Figure 5: Action-gap deviation as function of discount factor.

low discount factors. In short, $k$ controls the smallest Q-value that can still be accurately represented (i.e., for which the action gap in the log-space is still significant). Roughly, the smallest value that can still be accurately represented is about $\gamma^k$. In other words, the cut-off point lies approximately at a state from which it takes $k$ time steps to experience a +1 reward. Setting $k$ too high causes actions that have 0 value in the regular space to have a large negative value in the log-space. This can increase the action gap substantially for the corresponding states, thus, resulting in an overall increase of the action-gap deviation.

The parameters $c$ and $d$ scale and shift values in the logarithmic space and do not have any effect on the action-gap deviation. The parameter $d$ controls the initialization of the Q-values. Setting $d$ as follows:

$$d = -c \ln(q_{init} + \gamma^k), \tag{7}$$

ensures that $f^{-1}(0) = q_{init}$ for any value of $c, k$, and $\gamma$. This can be useful in practice, e.g., when using neural networks to represent $\widetilde{Q}$, as it enables standard initialization methods (which produce output values around 0) while still ensuring that the initialized $\widetilde{Q}$ values correspond with $q_{init}$ in the regular space. The parameter $c$ scales values in the log-space. For most tabular and linear methods, scaling values does not affect the optimization process. Nevertheless, in deep RL methods more advanced optimization techniques are commonly used and, thus, such scaling can impact the optimization process significantly. In all our experiments, except the deep RL experiments, we fixed $d$ according to the equation above with $q_{init} = 0$ and used $c = 1$.

In stochastic environments, the approach described in this section causes issues, because averaging over stochastic samples in the log-space produces an underestimate compared to averaging in the regular space and then mapping the result to the log-space. Specifically, if $X$ is a random variable, $\mathbb{E}\left[\ln(X)\right] \leq \ln\left(\mathbb{E}[X]\right)$ (i.e., Jensen's inequality). Fortunately, within our specific context, there is a way around this limitation that we discuss in the next section.

## 4.2  Stochastic Domains with Positive Rewards

The step-size $\alpha$ generally conflates two forms of averaging: averaging of stochastic update targets due to environment stochasticity, and, in the case of function approximation, averaging over different states. To amend our method for stochastic environments, ideally, we would separate these forms of averaging and perform the averaging over stochastic update targets in the regular space and the averaging over different states in the log-space. While such a separation is hard to achieve, the approach presented below, which is inspired by the above observation, achieves many of the same benefits. In particular, it enables convergence of $\widetilde{Q}$ to $f(Q^*)$, even when the environment is stochastic.

Let $\beta_{log}$ be the step-size for averaging in the log-space, and $\beta_{reg}$ be the step-size for averaging in the regular space. We amend the approach from the previous section by computing an alternative update target that is based on performing an averaging operation in the regular space. Specifically, the update target $U_t$ is transformed into an alternative update target $\hat{U}_t$ as follows:

$$\hat{U}_t := f^{-1}(\widetilde{Q}_t(s_t, a_t)) + \beta_{reg}\left(U_t - f^{-1}(\widetilde{Q}_t(s_t, a_t))\right), \tag{8}$$

with $U_t := r_t + \gamma \max_{a'} f^{-1}(\widetilde{Q}_t(s_{t+1}, a'))$. The modified update target $\hat{U}_t$ is used for the update in the log-space:

$$\widetilde{Q}_{t+1}(s_t, a_t) := \widetilde{Q}_t(s_t, a_t) + \beta_{log}\left(f(\hat{U}_t) - \widetilde{Q}_t(s_t, a_t)\right). \tag{9}$$

Note that if $\beta_{reg} = 1$, then $\hat{U}_t = U_t$, and update (9) reduces to update (5) from the previous section, with $\alpha = \beta_{log}$.

The conditions for convergence are discussed in the next section, but one of the conditions is that $\beta_{reg}$ should go to 0 in the limit. From a more practical point of view, when using fixed values for $\beta_{reg}$ and $\beta_{log}$, $\beta_{reg}$ should be set sufficiently small to keep underestimation of values due to the averaging in the log-space under control. To illustrate this, we plot the RMS error on a positive-reward variant of the chain task ($r_R = 0, r_L = +1, p = 0.25$). The RMS error plotted is based on the difference between $f^{-1}(\widetilde{Q}(s, a))$ and $Q^*(s, a)$ over all state-action pairs. We used a tile-width of 1, corresponding with

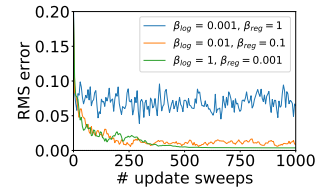

Figure 6: RMS error.

a tabular representation, and used $k = 200$. Note that for $\beta_{reg} = 1$, which reduces the method to the one from the previous section, the error never comes close to zero.

## 4.3 Stochastic Domains with Positive and/or Negative Rewards

We now consider the general case where the rewards can be both positive or negative (or zero). It might seem that we can generalize to negative rewards simply by replacing $x$ in the mapping function (6) by $x + D$, where $D$ is a sufficiently large constant that prevents $x + D$ from becoming negative. The problem with this approach, as we will demonstrate empirically, is that it does not decrease $\kappa$ for low discount factors. Hence, in this section we present an alternative approach, based on decomposing the Q-value function into two functions.

Consider a decomposition of the reward $r_t$ into two components, $r_t^+$ and $r_t^-$, as follows:

$$r_t^+ := \begin{cases} r_t & \text{if } r_t \geq 0 \\ 0 & \text{otherwise} \end{cases} \quad ; \quad r_t^- := \begin{cases} |r_t| & \text{if } r_t < 0 \\ 0 & \text{otherwise} \end{cases} . \tag{10}$$

Note that $r_t^+$ and $r_t^-$ are always non-negative and that $r_t = r_t^+ - r_t^-$ at all times. By decomposing the observed reward in this manner, these two reward components can be used to train two separate Q-value functions: $\widetilde{Q}^+$, which represents the value function in the mapping space corresponding to $r_t^+$, and $\widetilde{Q}^-$, which plays the same role for $r_t^-$. To train these value functions, we construct the following update targets:

$$U_t^+ := r_t^+ + \gamma f^{-1}\left(\widetilde{Q}_t^+(s_{t+1}, \tilde{a}_{t+1})\right) \quad ; \quad U_t^- := r_t^- + \gamma f^{-1}\left(\widetilde{Q}_t^-(s_{t+1}, \tilde{a}_{t+1})\right) \tag{11}$$

with $\tilde{a}_{t+1} := \arg\max_{a'}\left(f^{-1}(\widetilde{Q}_t^+(s_{t+1}, a')) - f^{-1}(\widetilde{Q}_t^-(s_{t+1}, a'))\right)$. These update targets are modified into $\hat{U}_t^+$ and $\hat{U}_t^-$, respectively, based on (8), which are then used to update $\widetilde{Q}^+$ and $\widetilde{Q}^-$, respectively, based on (9). Action-selection at time $t$ is based on $Q_t$, which we define as follows:

$$Q_t(s, a) := f^{-1}\left(\widetilde{Q}_t^+(s, a)\right) - f^{-1}\left(\widetilde{Q}_t^-(s, a)\right) \tag{12}$$

In the supplementary material, we prove convergence of logarithmic Q-learning under similar conditions as regular Q-learning. In particular, the product $\beta_{log,t} \cdot \beta_{reg,t}$ has to satisfy the same conditions as $\alpha_t$ does for regular Q-learning. There is one additional condition on $\beta_{reg,t}$, which states that it should go to zero in the limit.

We now compute $\kappa$ for the full version of the chain task. Because there are two functions, $\widetilde{Q}^+$ and $\widetilde{Q}^-$, we have to generalize the definition of $\kappa$ to this situation. We consider three generalizations: 1) $\kappa$ is based on the action-gaps of $\widetilde{Q}^+$ ('log plus-only'); 2) $\kappa$ is based on the action-gaps of $\widetilde{Q}^-$ ('log min-only'); and 3) $\kappa$ is based on the action-gaps of both $\widetilde{Q}^-$ and $\widetilde{Q}^+$ ('log both'). Furthermore, we plot a version that resolves the issue of negative rewards naïvely, by adding a value $D = 1$ to the input of the log-function ('log bias'). We plot $\kappa$ for these variants in Figure 7, using $k = 200$, together with $\kappa$ for regular Q-learning ('reg'). Interestingly, only for the 'log plus-only' variant $\kappa$ is small for all discount factors. Further analysis showed that the reason for this is that under the

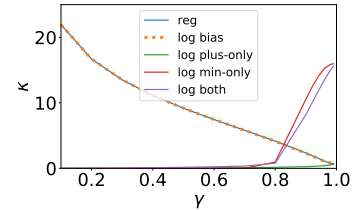

Figure 7: Action-gap deviation as function of discount factor.

optimal policy, the chance that the agent moves from a state close to the positive terminal state to the negative terminal state is very small, which means that $k = 200$ is too small to make the action-gaps for $\widetilde{Q}^-$ homogeneous. However, as we will see in the next section, the performance with $k = 200$ is good for all discount factors, demonstrating that not having homogeneous action-gaps for $\widetilde{Q}^-$ is not a huge issue. We argue that this could be because of the behavior related to the nature of positive and negative rewards: it might be worthwhile to travel a long distance to get a positive reward, but avoiding a negative reward is typically a short-horizon challenge.

# 5 Experiments

We test our method by returning to the full version of the chain task and the same performance metric $F$ as used in Section 3.2, which measures whether or not the greedy policy is optimal. We used $k = 200$, $\beta_{reg} = 0.1$, and $\beta_{log} = 0.01$ (the value of $\beta_{reg} \cdot \beta_{log}$ is equal to the value of $\alpha$ used in Section 3.2). Figure 8 plots the result for early learning as well as the final performance. Comparing these graphs with the graphs from Figure 3 shows that logarithmic Q-learning has successfully resolved the optimization issues of regular Q-learning related to the use of low discount factors in conjunction with function approximation.

Combined with the observation from Section 3.1 that the best discount factor is task-dependent, and the convergence proof in the supplementary material, which guarantees that logarithmic Q-learning converges to the same policy as regular Q-learning, these results demonstrate that logarithmic Q-learning is able to solve tasks that are challenging to solve with Q-learning. Specifically, if a finite-horizon performance metric is used and the task is such that the metric gap is substantially smaller for lower discount factors, but performance falls flat for these discount factors due to function approximation.

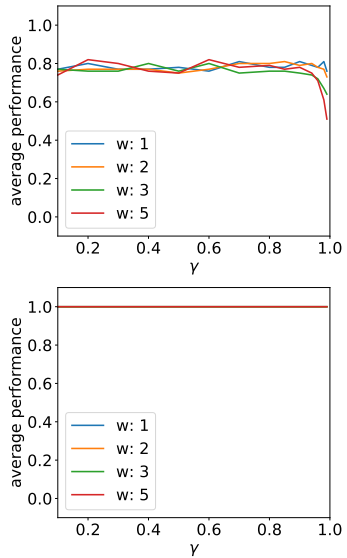

Figure 8: Early performance (top) and final performance (bottom) on the chain task.

Finally, we test our approach in a more complex setting by comparing the performance of DQN [Mnih et al., 2015] with a variant of it that implements our method, which we will refer to as LogDQN.[3]
To enable easy baseline comparisons, we used the Dopamine framework for our experiments [Castro et al., 2018]. This framework not only contains open-source code of several important deep RL methods, but also contains the results obtained with these methods for a set of 60 games from the Arcade Learning Environment [Bellemare et al., 2013, Machado et al., 2018]. This means that direct comparison to some important baselines is possible.

Our LogDQN implementation consists of a modification of the Dopamine's DQN code. Specifically, in order to adapt DQN's model to provide estimates of both $\widetilde{Q}^+$ and $\widetilde{Q}^-$, the final output layer is doubled in size, and half of it is used to estimate $\widetilde{Q}^+$ while the other half estimates $\widetilde{Q}^-$. All the other layers are shared between $\widetilde{Q}^+$ and $\widetilde{Q}^-$ and remain unchanged. Because both $\widetilde{Q}^+$ and $\widetilde{Q}^-$ are updated using the same samples, the replay memory does not require modification, so the memory footprint does not change. Furthermore, because $\widetilde{Q}^+$ and $\widetilde{Q}^-$ are updated simultaneously using a single pass through the model, the computational cost of LogDQN and DQN is similar. Further implementation details are provided in the supplementary material.

The published Dopamine baselines are obtained on a stochastic version of Atari using *sticky actions* [Machado et al., 2018] where with 25% probability the environment executes the action from the previous time step instead of the agent's new action. Hence, we conducted all our LogDQN experiments on this stochastic version of Atari as well.

While Dopamine provides baselines for 60 games in total, we only consider the subset of 55 games for which human scores have been published, because only for these games a 'human-normalized score' can be computed, which is defined as:

$$\frac{\text{Score}_{\text{Agent}} - \text{Score}_{\text{Random}}}{\text{Score}_{\text{Human}} - \text{Score}_{\text{Random}}}. \tag{13}$$

We use Table 2 from Wang et al. [2016] to retrieve the human and random scores.

We optimized hyper-parameters using a subset of 6 games. In particular, we performed a scan over the discount factor $\gamma$ between $\gamma = 0.84$ and $\gamma = 0.99$. For DQN, $\gamma = 0.99$ was optimal; for LogDQN, the best value in this range was $\gamma = 0.96$. We tried lower $\gamma$ values as well, such as $\gamma = 0.1$ and $\gamma = 0.5$, but this did not improve the overall performance over these 6 games. For the other hyper-parameters of LogDQN we used $k = 100$, $c = 0.5$, $\beta_{log} = 0.0025$, and $\beta_{reg} = 0.1$. The

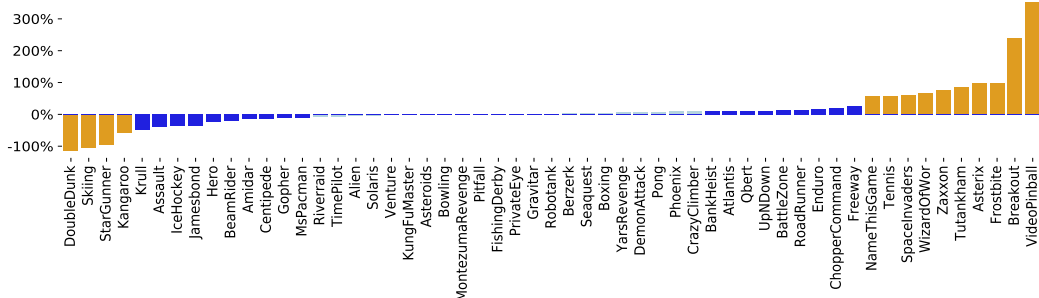

Figure 10: Relative performance of LogDQN w.r.t. DQN (positive percentage means LogDQN outperforms DQN). Orange bars indicate a performance difference larger than 50%; dark-blue bars indicate a performance difference between 10% and 50%; light-blue bars indicate a performance difference smaller than 10%.

product of $\beta_{log}$ and $\beta_{reg}$ is 0.00025, which is the same value as the (default) step-size $\alpha$ of DQN. We used different values of $d$ for the positive and negative heads: we set $d$ based on (7) with $q_{init} = 1$ for the positive head, and $q_{init} = 0$ for the negative head. Results from the hyper-parameter optimization, as well as further implementation details are provided in the supplementary material.

Figure 10 shows the performance of LogDQN compared to DQN per game, using the same comparison equation as used by Wang et al. [2016]:

$$\frac{\text{Score}_{\text{LogDQN}} - \text{Score}_{\text{DQN}}}{\max(\text{Score}_{\text{DQN}}, \text{Score}_{\text{Human}}) - \text{Score}_{\text{Random}}}.$$

where $\text{Score}_{\text{LogDQN/DQN}}$ is computed by averaging over the last 10% of each learning curve (i.e., last 20 epochs).

Figure 9 shows the mean and median of the human-normalized score of LogDQN, as well as DQN. We also plot the performance of the other baselines that Dopamine provides: C51 [Bellemare et al., 2017], Implicit Quantile Networks [Dabney et al., 2018], and Rainbow [Hessel et al., 2018]. These baselines are just for reference; we have not attempted to combine our technique with the techniques that these other baselines make use of.

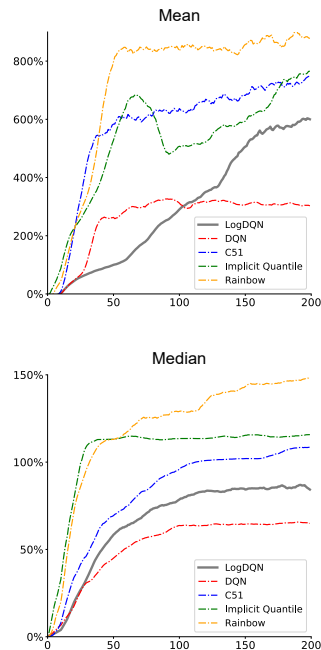

Figure 9: Human-normalized mean (left) and median (right) scores on 55 Atari games for LogDQN and various other algorithms.

## 6 Discussion and Future Work

Our results provide strong evidence for our hypothesis that large differences in action-gap sizes are detrimental to the performance of approximate RL. A possible explanation could be that optimizing on the $L_2$-norm (3) might drive towards an average squared-error that is similar across the state-space. However, the error-landscape required to bring the approximation error below the action-gap across the state-space has a very different shape if the action-gap is orders of magnitude different in size across the state-space. This mismatch between the required error-landscape and that produced by the $L_2$-norm might lead to an ineffective use of the function approximator. Further experiments are needed to confirm this.

The strong performance we observed for $\gamma = 0.96$ in the deep RL setting is unlikely solely due to a difference in metric gap. We suspect that there are also other effects at play that make LogDQN as effective as it is. On the other hand, at (much) lower discount factors, the performance was not as good as it was for the high discount factors. We believe a possible reason could be that since such low values are very different than the original DQN settings, some of the other DQN hyper-parameters might no longer be ideal in the low discount factor region. An interesting future direction would be to re-evaluate some of the other hyper-parameters in the low discount factor region.

**Acknowledgments**

We like to thank Kimia Nadjahi for her contributions to a convergence proof of an early version of logarithmic Q-learning. This early version ultimately was replaced by a significantly improved version that required a different convergence proof.

## Footnotes

[1]As an example, in the seminal work of Mnih et al. [2015], the (undiscounted) score of Atari games is reported with a time-limit of 5 minutes per game.

[2]This is the main reason why optimizing on an infinite-horizon objective, rather than a finite-horizon one, is an attractive choice.

[3]The code for the experiments can be found at: `https://github.com/microsoft/logrl`

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
