[Supplementary Material · NeurIPS_2019___supp_material_final.pdf]

# Using a Logarithmic Mapping to Enable Lower Discount Factors in Reinforcement Learning
## — *Supplementary Material* —

## A  Proof of Convergence for Logarithmic Q-learning

### A.1  Definitions and Theorem

Our logarithmic Q-learning method is defined by the following equations:

$$f(x) := c \ln(x + \gamma^k) + d \quad ; \quad f^{-1}(x) := e^{(x-d)/c} - \gamma^k \tag{1}$$

$$r_t^+ := \begin{cases} r_t & \text{if } r_t \geq 0 \\ 0 & \text{otherwise} \end{cases} \quad ; \quad r_t^- := \begin{cases} |r_t| & \text{if } r_t < 0 \\ 0 & \text{otherwise} \end{cases} \tag{2}$$

$$Q_t(s,a) := f^{-1}\left(\widetilde{Q}_t^+(s,a)\right) - f^{-1}\left(\widetilde{Q}_t^-(s,a)\right) \tag{3}$$

$$\tilde{a}_{t+1} := \arg\max_{a'}\left(Q_t(s_{t+1}, a')\right) \tag{4}$$

$$U_t^+ := r_t^+ + \gamma f^{-1}\left(\widetilde{Q}_t^+(s_{t+1}, \tilde{a}_{t+1})\right) \tag{5}$$

$$\hat{U}_t^+ := f^{-1}\left(\widetilde{Q}_t^+(s_t, a_t)\right) + \beta_{reg,t}\left(U_t^+ - f^{-1}\left(\widetilde{Q}_t^+(s_t, a_t)\right)\right) \tag{6}$$

$$\widetilde{Q}_{t+1}^+(s_t, a_t) := \widetilde{Q}_t^+(s_t, a_t) + \beta_{log,t}\left(f\left(\hat{U}_t^+\right) - \widetilde{Q}_t^+(s_t, a_t)\right) \tag{7}$$

$$U_t^- := r_t^- + \gamma f^{-1}\left(\widetilde{Q}_t^-(s_{t+1}, \tilde{a}_{t+1})\right) \tag{8}$$

$$\hat{U}_t^- := f^{-1}\left(\widetilde{Q}_t^-(s_t, a_t)\right) + \beta_{reg,t}\left(U_t^- - f^{-1}\left(\widetilde{Q}_t^-(s_t, a_t)\right)\right) \tag{9}$$

$$\widetilde{Q}_{t+1}^-(s_t, a_t) := \widetilde{Q}_t^-(s_t, a_t) + \beta_{log,t}\left(f\left(\hat{U}_t^-\right) - \widetilde{Q}_t^-(s_t, a_t)\right) \tag{10}$$

For these equations, the following theorem holds:

**Theorem 1** *Under the definitions above, $Q_t$ converges to $Q^*$ w.p. 1 if the following conditions hold:*

1. $0 \leq \beta_{log,t} \cdot \beta_{reg,t} \leq 1$
2. $\sum_{t=0}^{\infty} \beta_{log,t} \cdot \beta_{reg,t} = \infty$
3. $\sum_{t=0}^{\infty} (\beta_{log,t} \cdot \beta_{reg,t})^2 < \infty$
4. $\lim_{t \to \infty} \beta_{reg,t} = 0$

## A.2 Proof - part 1

We define $Q_t^+(s,a) := f^{-1}(\widetilde{Q}_t^+(s,a))$ and prove in part 2 that from (7), (6), and (5) it follows that:

$$Q_{t+1}^+(s_t, a_t) = Q_t^+(s_t, a_t) + \beta_{reg,t} \cdot \beta_{log,t} \left( U_t^+ - Q_t^+(s_t, a_t) + c_t^+ \right), \qquad (11)$$

with $c_t^+$ converging to zero w.p. 1 under condition 4 of the theorem, and $U_t^+$ defined as:

$$U_t^+ := r_t^+ + \gamma \, Q_t^+(s_{t+1}, \tilde{a}_{t+1}).$$

Similarly, using definition $Q_t^-(s,a) := f^{-1}(\widetilde{Q}_t^-(s,a))$ and (10), (9), and (8) it follows that:

$$Q_{t+1}^-(s_t, a_t) = Q_t^-(s_t, a_t) + \beta_{reg,t} \cdot \beta_{log,t} \left( U_t^- - Q_t^-(s_t, a_t) + c_t^- \right), \qquad (12)$$

with $c_t^-$ converging to zero w.p. 1 under condition 4 of the theorem, and $U_t^-$ defined as:

$$U_t^- := r_t^+ + \gamma \, Q_t^-(s_{t+1}, \tilde{a}_{t+1}).$$

It follows directly from the definitions of $Q_t^+$ and $Q_t^-$ and (3) that:

$$Q_t(s,a) = Q_t^+(s,a) - Q_t^-(s,a). \qquad (13)$$

Subtracting (12) from (11) and substituting this equivalence yields:

$$Q_{t+1}(s_t, a_t) = Q_t(s_t, a_t) + \beta_{reg,t} \cdot \beta_{log,t} \left( r_t^+ - r_t^- + \gamma Q_t(s_{t+1}, \tilde{a}_{t+1}) - Q_t(s_t, a_t) + c_t^+ - c_t^- \right).$$

From (2) it follows that $r_t = r_t^+ - r_t^-$. Furthermore, the following holds:

$$
\begin{aligned}
Q_t(s_{t+1}, \tilde{a}_{t+1}) &= Q_t \left( s_{t+1}, \arg\max_{a'} \left( Q_t(s_{t+1}, a') \right) \right) \\
&= \max_{a'} Q_t \left( s_{t+1}, a' \right)
\end{aligned}
$$

Using these equivalences and defining $\alpha_t := \beta_{reg,t} \cdot \beta_{log,t}$ and $c_t := c_t^+ - c_t^-$, it follows that:

$$Q_{t+1}(s_t, a_t) = Q_t(s_t, a_t) + \alpha_t \left( r_t + \gamma \max_{a'} Q_t(s_{t+1}, a') - Q_t(s_t, a_t) + c_t \right), \qquad (14)$$

with $c_t$ converging to zero w.p. 1 under condition 4 of the theorem. This is a noisy Q-learning algorithm with the noise term decaying to zero. As we show in part 2, $c_t$ is fully specified (in the positive case, and likewise in the negative case) by $Q_t^+$, $U_t^+$, and $\beta_{reg,t}$, which implies that $c_t$ is measurable given information at time $t$, as required by Lemma 1 in Singh et al. [2000]. Invoking that Lemma, it can therefore be shown that the iterative process defined by (14) converges to $Q_t^*$ if $0 \leq \alpha \leq 1$, $\sum_{t=0}^{\infty} \alpha = \infty$, and $\sum_{t=0}^{\infty} \alpha_t^2 < \infty$, as is guaranteed by the first three conditions of the theorem. The steps are similar to the proof of Theorem 1 of the same reference, which we do not repeat here.

## A.3 Proof - part 2

In this section, we prove that (11) holds under the definitions from Section A.1, $Q_t^+(s,a) := f^{-1}(\widetilde{Q}_t^+(s,a))$, and condition 4 of the theorem. The proof of (12) follows the same steps, but with the '-' variants of the different variables instead. For readability, we use $\beta_1$ for $\beta_{log,t}$ and $\beta_2$ for $\beta_{reg,t}$.

The definition of $Q_t^+$ implies $\widetilde{Q}_t^+(s,a) = f\left(Q_t^+(s,a)\right)$. Using these equivalences, we can rewrite (7), (6), and (5) in terms of $Q_t$:

$$f(Q_{t+1}^+(s_t, a_t)) = f(Q_t^+(s_t, a_t)) + \beta_1 \left( f(\hat{U}_t^+) - f(Q_t^+(s_t, a_t)) \right), \qquad (15)$$

with

$$\hat{U}_t^+ = Q_t^+(s_t, a_t) + \beta_2 \left( r_t^+ + \gamma \, Q_t^+(s_{t+1}, \tilde{a}_{t+1}) - Q_t^+(s_t, a_t) \right). \qquad (16)$$

By applying $f^{-1}$ to both sides of (15), we get:

$$Q_{t+1}^+(s_t, a_t) = f^{-1} \left( f(Q_t^+(s_t, a_t)) + \beta_1 \left( f\left(\hat{U}_t^+\right) - f(Q_t^+(s_t, a_t)) \right) \right), \qquad (17)$$

which can be rewritten as:

$$Q_{t+1}^+(s_t, a_t) = Q_t^+(s_t, a_t) + \beta_1 \left( \hat{U}_t^+ - Q_t^+(s_t, a_t) \right) + e_t^+, \qquad (18)$$

where $e_t^+$ is the error due to averaging in the log-space instead of in the regular space:

$$e_t^+ := f^{-1}\left(f(Q_t^+(s_t, a_t)) + \beta_1\left(f(\hat{U}_t^+) - f(Q_t^+(s_t, a_t))\right)\right)$$

$$- Q_t^+(s_t, a_t) - \beta_1\left(\hat{U}_t^+ - Q_t^+(s_t, a_t)\right) \quad (19)$$

The key to proving (11), and by extension the theorem, is proving that $e_t^+$ goes sufficiently fast to 0. We prove this by defining a bound on $|e_t^+|$ and showing that this bound goes to 0. Figure 1 illustrates the bound. The variables in the figure refer to the following quantities:

$$
\begin{aligned}
a &\to Q_t^+(s_t, a_t) \\
b &\to \hat{U}_t^+ \\
v &\to (1 - \beta_1)\,a + \beta_1\,b \\
\tilde{w} &\to (1 - \beta_1)f(a) + \beta_1 f(b) \\
w &\to f^{-1}(\tilde{w})
\end{aligned}
$$

The error $e_t^+$ corresponds with:

$$e_t^+ = f^{-1}\big((1 - \beta_1)f(a) + \beta_1 f(b)\big) - \big((1 - \beta_1)a + \beta_1 b\big) = f^{-1}(\tilde{w}) - v = w - v$$

Note here that since $f$ is a strictly concave function, the definition of $\tilde{w}$ and $v$ directly imply $\tilde{w} < f(v)$. Because $f^{-1}$ is monotonically increasing, it follow that $w < v$, which yields $|e_t^+| = v - w$.

Figure 1: Bounding the error, for the case $a < b$ (left) and for $a > b$ (right).

In both graphs of Figure 1, besides the mapping function $f(x)$, three more functions are plotted: $g_0(x)$, $g_1(x)$, and $g_2(x)$. These three functions are all linear functions passing through the point $(a, f(a))$. The function $g_0(x)$ has derivative $f'(a)$, while $g_2(x)$ has derivative $f'(b)$. The function $g_1(x)$ passes through point $(b, f(b))$ as well, giving it derivative $(f(a) - f(b))/(a - b)$.

As illustrated by the figure, $g_1(v) = \tilde{w}$ and $g_1^{-1}(\tilde{w}) = v$. Furthermore, for $x$ between $a$ and $b$ the following holds (in both cases):

$$g_0(x) \geq f(x) \geq g_1(x) \geq g_2(x)$$

And, equivalently:

$$g_0^{-1}(x) \leq f^{-1}(x) \leq g_1^{-1}(x) \leq g_2^{-1}(x)\,.$$

We bound $|e_t^+| = v - w$, by using a lowerbound $w^-$ for $w$ and an upperbound $v^+$ for $v$. Specifically, we define $w^- := g_0^{-1}(\tilde{w})$ and $v^+ := g_2^{-1}(\tilde{w})$, and can now bound the error as follows: $|e_t^+| <= v^+ - w^-$. Next, we compute an expression for the bound in terms of $a$, $b$, and $f$.

First, note that for the derivatives of $g_0$ and $g_2$ the following holds:

$$g_0'(x) = f'(a) = \frac{f(a) - \tilde{w}}{a - w^-} \quad ; \quad g_2'(x) = f'(b) = \frac{f(a) - \tilde{w}}{a - v^+}\,.$$

From this it follows that:

$$w^- = \frac{\tilde{w} - f(a)}{f'(a)} + a \quad ; \quad v^+ = \frac{\tilde{w} - f(a)}{f'(b)} + a .$$

Using this, we rewrite our bound as:

$$
\begin{aligned}
v^+ - w^- &= \frac{\tilde{w} - f(a)}{f'(b)} - \frac{\tilde{w} - f(a)}{f'(a)} \\
&= \left( \frac{1}{f'(b)} - \frac{1}{f'(a)} \right) \cdot (\tilde{w} - f(a)) \\
&= \left( \frac{1}{f'(b)} - \frac{1}{f'(a)} \right) \cdot \Big( (1 - \beta_1) f(a) + \beta_1 f(b) - f(a) \Big) \\
&= \beta_1 \left( \frac{1}{f'(b)} - \frac{1}{f'(a)} \right) \Big( f(b) - f(a) \Big)
\end{aligned}
$$

Recall that $f(x) := c \ln(x + \gamma^k) + d$. The derivative of $f(x)$ is

$$f'(x) = \frac{c}{x + \gamma^k}$$

Substituting $f(x)$ and $f'(x)$ in the expression for the bound gives:

$$
\begin{aligned}
v^+ - w^- &= \beta_1 \left( \frac{b + \gamma^k}{c} - \frac{a + \gamma^k}{c} \right) \Big( c \ln(b + \gamma^k) + d - (c \ln(a + \gamma^k) + d) \Big) \\
&= \beta_1 (b - a)(\ln(b + \gamma^k) - \ln(a + \gamma^k)) \\
&= \beta_1 (a - b)(\ln(a + \gamma^k) - \ln(b + \gamma^k)) \\
&= \beta_1 (a - b) \ln \left( \frac{a + \gamma^k}{b + \gamma^k} \right) \\
&= \beta_1 (a - b) \ln \left( \frac{a - b}{b + \gamma^k} + 1 \right)
\end{aligned}
$$

Using the definitions of $a$ and $b$, the results for the bound for $e_t^+$:

$$|e_t^+| \le v^+ - w^- \le \beta_1 (Q_t^+(s_t, a_t) - \hat{U}_t^+) \ln \left( \frac{Q_t^+(s_t, a_t) - \hat{U}_t^+}{\hat{U}_t^+ + \gamma^k} + 1 \right) \tag{20}$$

Definition (6) can be written as:

$$\hat{U}_t^+ := Q_t^+(s_t, a_t) + \beta_{reg,t} \left( U_t^+ - Q_t^+(s_t, a_t) \right) \tag{21}$$

yielding:

$$
\begin{aligned}
Q_t^+(s_t, a_t) - \hat{U}_t^+ &= Q_t^+(s_t, a_t) - \Big( Q_t^+(s_t, a_t) + \beta_{reg,t} \left( U_t^+ - Q_t^+(s_t, a_t) \right) \Big) \\
&= \beta_2 \left( Q_t^+(s_t, a_t) - U_t^+ \right)
\end{aligned}
$$

Substituting this in (20) gives:

$$|e_t^+| \le \beta_1 \beta_2 \big( Q_t^+(s_t, a_t) - U_t^+ \big) \ln \left( \frac{\beta_2 \left( Q_t^+(s_t, a_t) - U_t^+ \right)}{\hat{U}_t^+ + \gamma^k} + 1 \right)$$

Let us define $c_t^+$ as:

$$c_t^+ := \left( Q_t^+(s_t, a_t) - U_t^+ \right) \ln \left( \frac{\beta_2 \left( Q_t^+(s_t, a_t) - U_t^+ \right)}{\hat{U}_t^+ + \gamma^k} + 1 \right)$$

Hence, $|e_t^+| \le \beta_1 \beta_2 c_t^+$. Substituting maximum bound of $|e_t^+|$ and (21) in (18), we get:

$$Q_{t+1}^+(s_t, a_t) = Q_t^+(s_t, a_t) + \beta_1 \beta_2 \big( U_t^+ - Q_t^+(s_t, a_t) + c_t^+ \big) \tag{22}$$

with $c_t^+$ going to 0, if $\beta_2$ goes to 0, which concludes part 2 of the proof.

# B   Hypothesis Testing

The following hypotheses are tested: 1) lower discount factors cause poor performance because they result in smaller action gaps; 2) lower discount factors cause poor performance because they result in smaller relative action gaps (i.e, the action gap of a state divided by the maximum action-value of that state).

To test the first hypothesis, we performed the same experiment as in Section 3.2, but with rewards that are a factor 100 larger. This in turn increases the action gaps by a factor 100 as well. Hence, to validate the first hypothesis, this modification should improve (i.e., lower) the threshold value where the performance falls flat. To test the second hypothesis, we pushed all action-values up by 100 through additional rewards, reducing the relative action-gap. Specifically, the extra reward upon transitioning to a non-terminal state $100 \cdot (1 - \gamma)$, while the extra reward upon transition to a terminal state is 100. This effectively pushes all action-values up by exactly 100. To validate the second hypothesis, performance should degrade for this variation. We plotted the performance of these task variations, together with the performance on the regular task, in Figure 2. Both variations show roughly the same performance as the performance on the regular tasks, invalidating both hypotheses.

Figure 2: Performance on 3 variations of the chain task. Left: performance on the regular version; middle: performance on the variant with values 100 times larger; right: performance on the variant with values pushed up by 100. All versions result in roughly the same performance curves.

# C   Additional Details for Logarithmic DQN

In this section, we describe additional details specific to our deep RL (Atari) experiments.

## C.1   Implementation

In order to support reproducibility and enable reliable and accessible baseline comparisons, we base our implementation upon the Google's Dopamine framework [Castro et al., 2018]. Dopamine provides reliable, open-source code for several important deep RL algorithms (including DQN) and enables standardized benchmarking, yielding 'apples to apples' comparison under best known evaluation practices in RL. Therefore, we evaluate LogDQN without modifications of agent or environment parameters (w.r.t. those outlined by Castro et al. [2018]), except for LogDQN's hyper-parameters (i.e., $\gamma, k, c, \beta_{log}, \beta_{reg}$, and $d$; for which the chosen values are stated in the paper).

We now highlight any settings in our LogDQN implementation that differs from our formulation of the logarithmic Q-learning update rules, as follows:

- The most commonly-used loss function for DQN (and the default setting in Dopamine) is based on the Huber loss function [Huber, 1992], which slightly differs from the squared-error

loss specified as the general setting. While our results are for the standard Huber loss setting, in our primary experiments we did not observe any significant difference between the two.

- To optimize the loss function, we use the standard RMSProp optimizer[1] (as the default setting in Dopamine's DQN). This choice differs slightly from our logarithmic Q-learning formulation which illustrates the case for the fundamental gradient descent method.

- To initialize the LogDQN network, we generally use the standard Xavier initialization [Glorot and Bengio, 2010] scheme (also a Dopamine's default setting), with the mere exception of initializing the output-layer weights of our $Q^-$ function to zero (instead of small, noisy values around zero).

- We replaced the additive $\gamma^k$ in our original formulation of the mapping function, its inverse, and $d$ hyper-parameter with a minimum-clipping at $\gamma^k$ (i.e., enforcing the aforementioned value to be the minimum possible value in the corresponding computations). This gives a hard bound on the values that can be represented, instead of a soft bound, and increases the independence between the $k$ and $\gamma$ parameters, which is useful when optimizing hyper-parameters.

### C.2 Hyper-parameter tuning

The hyper-parameters of LogDQN used for the experiments are the result from an earlier hyper-parameter optimization performed using an older version of LogDQN that did not have a strategy to deal with stochastic environments (as described in Section 4.2). Due to time-constraints, we were unable to perform a new hyper-parameter optimization for the full version of LogDQN.

This earlier hyper-parameter optimization was performed across these 6 games: ALIEN, ZAXXON, BREAKOUT, DOUBLEDUNK, SPACEINVADERS, and FISHINGDERBY. For the discount factor, we tried $\gamma \in \{0.84, 0.92, 0.96, 0.98, 0.99\}$ and for $c$ we tried $c \in \{0.1, 0.5, 1.0, 2.0, 5.0\}$. Furthermore, $k$ was fixed at 100. For DQN, we tried the same $\gamma$ values. Figure 3 shows the mean and median human-normalized score across these 6 games.

In Figure 3, for LogDQN, the performance at the best $c$-value is plotted for each $\gamma$. The best values for LogDQN are $\gamma = 0.96$ and $c = 0.5$; for DQN, the best value is $\gamma = 0.99$ (according to the more robust median metric).

Figure 3: Mean and median performance across 6 games for (an incomplete version of) LogDQN and DQN.

## Footnotes

[1]See http://www.cs.toronto.edu/~tijmen/csc321/slides/lecture_slides_lec6.pdf

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

Figure 4: Learning curves for all 55 games.

Xavier Glorot and Yoshua Bengio. Understanding the difficulty of training deep feedforward neural networks. In *Proceedings of the 13th International Conference on Artificial Intelligence and Statistics*, pages 249–256, 2010.