[Reviews · NeurIPS 2019]

Reviewer 1



I thank the author's for their response. I would like to reemphasize that I like this paper a lot, and applaud the authors for their work. I strongly suggest an oral accept for this paper. Additionally, I've raised my score 1 point. However, I still felt that Section 3.1 was somewhat contrived, and stand by my initial criticism of this section. While it is an interesting example, and does demonstrate a particular problem that can emerge when the training metric differs from the performance metric, I still did not feel that it exactly demonstrated the point the authors were trying to make. ______________ Overall, I really like this paper. The authors introduce an evaluate a novel hypothesis (related to action-gaps) and they propose a natural remedy. The paper is very well written, and each topic is clearly presented and explained in a logical order. Overall, the points the authors make are convincing. Its worth noting that this work is particularly relevant in sparse-reward problems. In this setting, the advantage of store the value function in a logarthmic space is particularly clear, as it becomes feasible to learn the difference between the expected value of a single reward received in the future without becoming swamped by rewards received in the near-term. Sparse-reward problems are currently popular, so addressing a problem in this space seems valuable. While the authors also provide the standard Atari evaluations, they perhaps more importantly provide direct evidence for their hypothesis, while refuting the "action gap" hypothesis. This was a nice inclusion, due to the well-known problems with direct evaluation of methods in the field. There were some parts of the paper that were not totally convincing. I took some issue with the examples given in Figure 1, in particular with respect to the use of a finite horizon. Typically, the fixed horizon is applied during both training time and test time. In the example given by the author, the agent is allowed to run until it reaches a terminal state during training time, but during test time is cut off after h timesteps. The authors contrived an example wherein this results in an agent taking a path that is too long in some settings, meaning that the agent fails to reach the goal when larger discount factors are used. I think the purpose was to motivate the use of smaller discount factors in certain settings, but I did not find that this example really captures the reasons why they are useful. However, this wasn't a huge part of the paper, and thinking about the examples raised by the authors led to some useful insight that helped with understanding the rest of the paper even though I did not find the particular example convincing. The authors chose to apply their technique to generate a particular variant of Q-learning. It seems to me that value methods have fallen out of favor relative to policy gradient methods, but it is perfectly clear how to generalize the author's approach. Perhaps it would be worth a mention by the authors? The approach of splitting the rewards into positive and negative rewards is a litle hack-ish, but its not unreasonable in most problems, in particular with respect to sparse rewards problem. Their approach was marginally better than DQN on most Atari games, but then again almost everything beats vanilla DQN. Nevertheless, the comparison was fair. That being said, it would be nice to see some demonstrations where the advantage of the author's method was clear and convincing, even in a contrived setting. Overall, in spite of a few minor minor issues the paper was well-written, discussed a relevent problem, and introduced a novel fix. I would argue for the acceptance of this paper.

Reviewer 2



In my opinion, this is an excellent paper. It treats a highly interesting, general, important, and unsolved problem. The analysis, ideas, and algorithm are insightful. The paper is clearly written; the formalism is precise and well done, and kept at an appropriate and understandable level. I believe the paper is foundational. Originality: High Quality: High Clarity: High (see minor notes under improvements) Significance: High Response to author feedback: I'm in agreement with the authors on their response.

Reviewer 3



The authors provided thorough and detailed analysis of the discount factor's impact on the reinforcement learning process. Building upon their empirical observation, they contributed to a new metric, the standard deviation of the action-gap over sampled states, that better explained the impact of smaller discount factor on the optimization processes compared to previous metrics (smaller action gap or smaller relative action gap). They also proposed a principled logarithmic Q-Learning variant that allows lower discount factor in reinforcement learning optimization process by mapping the values to a logarithmic space and performing updates in the logarithmic space. They empirically show (1) that the proposed algorithm is able to mitigate the issues caused by small discount factor in regular Q-Learning and (2) that the proposed algorithm could also outperform DQN on six Atari games (namely Asterix, Breakout, Zaxxon, DoubleDunk, FishingBeDerby and Tennis). The paper is well-written and easy to follow. Some comments: -- (1) One question about the curves of A and B in Figure 1: shouldn't the performance change suddenly at a certain gamma value (for example around 0.725 for A)? But the plotted curves somehow look like smoothed''. Please let me know if I missed something. ==> Addressed in the authors' rebuttal. It is an artifact of their plot procedure. Will be updated in the final version. -- (2) Just out of curiosity, what would the learning curves look like for DQN with gamma=0.96? Also is logDQN more robust compared to the DQN baseline for different gamma values (in the investigated interval [0.84, 0.99])? It is of great interest to see the performance as a function of different gamma values for logDQN and DQN. ==> The authors made a good point that the interplay of different hyper-parameters of DQN makes this unrealistic. -- (3) Since the logDQN also changed the network architecture by adding more output units, the empirical result would be more significant if comparison against Dueling DQN [1] is provided. It is not necessary but it can make the results stronger. -- (4) Some previous work also discussed the usefulness of a smaller discount factor in planning setting when model errors are presented [2]. The studied problem is not the same but I think it is relevant to the paper in a sense that both papers provide some insights into the problem of selecting a gamma value for best evaluation performance. -- (5) minor typos: line265, movies [1] Wang, Ziyu, et al. "Dueling network architectures for deep reinforcement learning." arXiv preprint arXiv:1511.06581(2015). [2] Jiang, N., Kulesza, A., Singh, S., & Lewis, R. (2015, May). The dependence of effective planning horizon on model accuracy. In Proceedings of the 2015 International Conference on Autonomous Agents and Multiagent Systems (pp. 1181-1189). International Foundation for Autonomous Agents and Multiagent Systems.

[Author Response · NeurIPS 2019]

We thank all reviewers for their time and appreciate the thoughtful feedback. Below, we address the main comments.

**Reviewer 1:** *"In the example given by the author, the agent is allowed to run until it reaches a terminal state during training time, but during test time is cut off after h timesteps."*

We understand why this would be a concern, but it is actually not what we do. First, note that Figure 1 simply plots the optimal policies, which are theoretical entities that are independent of the training procedure (we computed them analytically). That being said, there is no reason why these optimal policies could not be learned with finite-length training episodes, all starting from the same initial state. As an example, consider that we want to learn $\pi_l^*$ using training episodes of length 12 (the same length as the performance metric). For simplicity, consider that we use Q-learning with a behavior policy that selects actions uniformly at random. Under this behavior policy there is a non-zero probability for each state-action pair that it will be visited within a single training episode. Hence, sufficient exploration occurs to enable convergence in the limit. A key detail to achieve convergence is that the moment the training episode reaches its final time step, this is not treated as a terminal state, but normal bootstrapping is used.

On the topic of terminal states, note that we have not explicitly defined any terminal states for the tasks from Figure 1. It might seem logical to define the state for which all positive objects are collected to be terminal, but this is not strictly necessary (it does not affect the optimal policy, nor the metric gap, nor the ability to learn these). We consider terminal states to be part of the MDP definition (see line 49). In other words, they are independent of the performance metric F or learning metric $F_l$. A finite performance metric does not introduce terminal states, just like stopping training after a finite number of steps does not introduce a terminal state. We will clarify this point further in the paper.

**R1:** *"Their approach was marginally better than DQN on most Atari games [...] it would be nice to see some demonstrations where the advantage of the author's method was clear and convincing [...]"*

We hope that our clarification of the Figure 1 plots has increased your appreciation of low discount factors. There is likely room for further improvement of the non-linear results, but we also believe that our current results show a decent improvement. Especially, because our technique is very general and can likely be combined with other techniques for improving performance. Furthermore, we argue—as you mention in your review as well—that the main contribution of this paper is more fundamental than the (obligatory) Atari evaluations.

**R1:** *"It seems to me that value methods have fallen out of favor relative to policy gradient methods, but it is perfectly clear how to generalize the author's approach. Perhaps it would be worth a mention by the authors?"*

Our approach can indeed be easily combined with policy-gradient methods. We will make a mention of this in the next version of the paper.

**Reviewer 2:** Thanks you very much for the kind words. We are delighted that you enjoyed reading it!

**Reviewer 3:** *"[...] the plotted curves somehow look like 'smoothed'. Please let me know if I missed something."*

This is due to an artifact of our plotting routine; in reality there is indeed a sudden change. We will update our plotting routine to remove this apparent smoothing.

**R3:** *"[...] is logDQN more robust compared to the DQN baseline for different gamma values (in the investigated interval [0.84, 0.99])? It is of great interest to see the performance as a function of different gamma values for logDQN and DQN."*

Generally spoken, applying our logarithmic mapping increases the robustness with respect to $\gamma$. This can be observed for the linear case by comparing Figures 3 and 8. However, properly evaluating the $\gamma$-dependent behavior for the non-linear case is non-trivial. The main reason for this is that DQN contains a lot of hidden hyper-parameters that work well for $\gamma = 0.99$, but it's unclear if these are also a good choice for different $\gamma$-values. As we mention in the Discussion section, we suspect that the reason that the optimal $\gamma$ we found for logDQN ($\gamma = 0.96$) is very close to the one for DQN ($\gamma = 0.99$) is related to this. We do plan to further investigate the non-linear setting in the future, including an in-dept evaluation of the $\gamma$-dependence between logDQN and DQN, but it's outside the scope of the current paper.

**R3:** *"Since the logDQN also changed the network architecture by adding more output units, the empirical result would be more significant if comparison against Dueling DQN is provided."*

We argue that a comparison of logDQN with Dueling DQN would not be as meaningful as our current comparison against DQN, because Dueling DQN does not just increase the output units, it is a different method altogether that enables generalization across actions. Furthermore, our logarithmic mapping approach is a general approach that can be combined with many existing methods, including Dueling DQN.

[Meta-Review · NeurIPS 2019]

The authors propose remapping value functions into a logarithmic space, leading to "logarithmic Q-learning" which is demonstrated to perform quite well in practice. This paper has by far the strongest overall scores (9, 9, 8) in my paper batch. All three reviewers are enthusiastic about the paper and its contributions and results. I am recommending that NeurIPS accept the paper for Oral presentation.